# In Silico Identification and Characterization of circRNAs as Potential Virulence-Related miRNA/siRNA Sponges from *Entamoeba histolytica* and Encystment-Related circRNAs from *Entamoeba invadens*

**DOI:** 10.3390/ncrna8050065

**Published:** 2022-09-26

**Authors:** Mario Ángel López-Luis, Cristian Julio César Padrón-Manrique, Jesús Alberto García-Lerena, Daniela Lozano-Amado, Rosaura Hernández-Rivas, Odila Saucedo-Cárdenas, Alfonso Méndez-Tenorio, Jesús Valdés

**Affiliations:** 1Laboratorio de Biotecnología y Bioinformática Genómica, Departamento de Bioquímica, Escuela Nacional de Ciencias Biológicas IPN Campus Lázaro Cárdenas, Mexico City 11340, Mexico; 2Departamento de Bioquímica, CINVESTAV-México, Av. IPN 2508 colonia San Pedro Zacatenco, GAM, CDMX, Mexico City 07360, Mexico; 3Departamento de Biomedicina Molecular, CINVESTAV-México, Mexico City 07360, Mexico; 4Departamento de Histología, Facultad de Medicina, Universidad Autónoma de Nuevo León, Monterrey 67700, Mexico

**Keywords:** CIRIfull, back splice junction, reverse overlap, differential expression, EIcirRNA, flicRNA, parasite, virulence

## Abstract

Ubiquitous eukaryotic non-coding circular RNAs regulate transcription and translation. We have reported full-length intronic circular RNAs (flicRNAs) in *Entamoeba histolytica* with esterified 3′ss and 5′ss. Their 5′ss GU-rich elements are essential for their biogenesis and their suggested role in transcription regulation. Here, we explored whether exonic, exonic-intronic, and intergenic circular RNAs are also part of the *E. histolytica* and *E. invadens* ncRNA RNAome and investigated their possible functions. Available RNA-Seq libraries were analyzed with the CIRI-full software in search of circular exonic RNAs (circRNAs). The robustness of the analyses was validated using synthetic decoy sequences with bona fide back splice junctions. Differentially expressed (DE) circRNAs, between the virulent HM1:IMSS and the nonvirulent Rahman *E. histolytica* strains, were identified, and their miRNA sponging potential was analyzed using the intaRNA software. Respectively, 188 and 605 reverse overlapped circRNAs from *E. invadens* and *E. histolytica* were identified. The sequence composition of the circRNAs was mostly exonic although different to human circRNAs in other attributes. 416 circRNAs from *E. histolytica* were virulent-specific and 267 were nonvirulent-specific. Out of the common circRNAs, 32 were DE between strains. Finally, we predicted that 8 of the DE circRNAs could function as sponges of the bioinformatically reported miRNAs in *E. histolytica*, whose functions are still unknown. Our results extend the *E. histolytica* RNAome and allow us to devise a hypothesis to test circRNAs/miRNAs/siRNAs interactions in determining the virulent/nonvirulent phenotypes and to explore other regulatory mechanisms during amoebic encystment.

## 1. Introduction

From their discovery 45 years ago [1,2], the widely conserved circular transcripts or circRNAs have been extensively studied, displaying important features such as exonuclease-resistance and ability to interact with proteins and other RNAs [3]. CircRNA molecules can be monoexonic, multi-exonic, exon-intronic, or intronic generated by co-transcriptional backsplicing, in which an upstream 3′ss acceptor is linked to a downstream 5´ss donor producing exonic circular molecules. Skipped exons within a lariat produced from linear splicing resulting may undergo post-transcriptional backsplicing originating exon-intron circRNAs (EIciRNAs) [4,5,6].

Backsplicing is also activated by splicing proteins and *cis*-elements. Alu repeats in flanking introns facilitate circularization by looping out of one or more exons [7,8], and CRISPR/Cas9-mediated Alu repeats gene editing prevents circularization [9]. In addition, the RNA binding proteins (RBP) FUS, and the splicing factors QKI and MBL have been associated with the circRNAs biogenesis [10,11,12]. Furthermore, intronic circRNAs are covalently closed lariat products resulting from processed intron lariats [6,13].

In keeping with their localization, circRNAs have different functions. Those with exonic sequences localize mainly to the cytoplasm and perform regulatory functions such as microRNAs and protein sponging, translation regulation, mRNA stability, and 5′ cap-independent translation [10,14,15,16,17,18]. Intronic circRNAs localize heterogeneously within the cell. In the nucleus, they are linked to gene expression regulation, while the function of cytoplasmatic intron circles remains unknown [19,20]. Some nuclear EIciRNAs are part of the RNA polymerase II (Pol II) binding complex able to downregulate their parental genes [4,21]. This is the case of the human 2′–5′ intronic circRNAs (ciRNAs), ci-ankrd52, and ci-sirt7, which directly bind to the serine 2-phosphorylated C-terminus domain (CTD) of Pol II, indicating that ciRNA-CTD interaction occurs during transcription elongation [21].

Shortly following its release from the spliceosome, a lariat becomes a circular intronic molecule after trimming off the 3´-end tail and before it is hydrolyzed by the debranching enzyme (Dbr1). However, numerous introns show cis-elements that confer them stability and accumulate inside the cell, like the stable sequence introns (sisRNAs) [19]. SisRNAs include 2′–5′ covalently linked intronic circular RNAs, which are basically processed lariats, and 3′–5′ covalently linked species with untrimmed 3´-end tails.

*Entamoeba histolytica* is the causative agent of amebiasis, a neglected tropical disease that still accounts for 55,500 deaths and nearly 2.4 million disability-adjusted life years [22]. Much effort has been devoted to understanding the mechanisms leading to infection establishment and development of disease and to treating such infections as well. In search of alternative theragnostic target molecules, we recently identified full-length intronic circular RNAs (flicRNAs) in the human protozoan parasite *E. histolytica* [23]. In agreement with previous reports [24], the 5′ss and 3′ss ends of these molecules are covalently linked with an average of 10 nucleotides between the branch point (BP) and the 3´ss in their lariat precursors [23]. We observed that flicRNAs are post-splicing products originating from the lariat by mechanisms not fully understood. In vivo splicing assays showed that the conserved GU-rich 5′ss is required for flicRNAs biogenesis. Furthermore, mutation of such elements decreased flicRNAs production and increased expression of their parental genes, indicating their role in the gene expression regulation [23]. Strikingly, although circular RNAs are expressed throughout the biota, they have been reported in a small number of protozoans [25]. Notwithstanding, other non-coding (nc) RNAs are expressed in Amoebozoa. For example, although not fully characterized, Mar-Aguilar et al. [26] confirmed the presence of microRNA-like molecules that had been predicted bioinformatically in *E. histolytica* [27]. miRNAs have been identified in *Dictyostelium discoideum* also [28]. Other ncRNAs in *Entamoeba* are the stress-induced self-circularized 5′-external transcribed spacer rRNAs in *E. histolytica* [29] and the encystation-related long ncRNAs in *E. invadens* [30].

To cope with protist and bacterial infections, recent works proposed that hosts’ immune response trigger circRNAs–miRNAs–mRNA regulatory networks [31,32,33]. However, little is known about whether these regulatory networks exist in *E. histolytica* and if they enable the onset of parasitic infections and invasive diseases. To address this question, we mined available RNA-seq libraries in search of circular exonic transcripts with miRNA sponging potential differentially expressed (DE) in HM1:IMSS (virulent) and Rahman (avirulent) *E. histolytica* strains. We also searched for circular transcripts in the reptile parasite *E. invadens*. After validation of the software using synthetic decoy sequences with *bona fide* back splice junctions (BSJ), we identified 188 and 605 reverse overlapped (RO) circRNAs from the *E. invadens* and *E. histolytica* libraries, respectively. The sequence composition of the identified circRNAs was mostly exonic. A total of 416 circRNAs from *E. histolytica* were virulent-specific and 267 were avirulent-specific. In addition, out of the common circRNAs, 32 were DE between strains, showing no correlation between intron length or the number of introns per locus with the number of circRNAs produced in a given locus. Finally, we predicted that eight of the DE circRNAs could function as sponges of the reported miRNAs in *E. histolytica*, whose functions are still unknown. Our results allow us to devise a working hypothesis to test the relationships between circRNAs and miRNAs in determining virulent/nonvirulent phenotypes and to explore additional regulatory mechanisms during amoebic trophozoite to cyst differentiation.

## 2. Results

### 2.1. Computational Identification of Entamoeba circRNAs

CircRNAs were identified based on backspliced RO reads as described previously [34] (Appendix A depicts the workflow used here). To identify the circRNAs in *E. histolytica* and *E. invadens*, available RNAseq libraries [35,36] were mined using the CIRI-full package. Respectively, six *E. histolytica* (run access codes ERR058005–ERR058007 for the virulent HM1:IMSS strain and ERR058008–ERR058010 for the avirulent Rahman strain) and four *E. invadens* (run access codes GSM4108195–GSM4108198) libraries were used, and the access codes for the reference genomes were GCF_000208925 and GCA_000330505.

Despite the robustness of the CIRI-full package [34] to ensure that it could identify circRNAs derived from an AT-rich genome such as *E. histolytica*, we performed a prospective run in a fraction of the libraries and found a putative circRNA from locus EHI_110790. Next, we used the exon end sequences as back splice junctions and manually added random sequences from the same locus to obtain 10 circular RNA decoys with displaced backsplice junctions. These sequences were incorporated into the partial libraries, and circRNAs were re-identified. In addition to the previous results, all decoy sequences were rescued in the analysis (Appendix A).

### 2.2. E. histolytica circRNAs

In *E. histolytica*, 958 circRNAs were identified. After filtering for duplicates, 605 circRNAs remained. Of these, 433 were found in the HM1:IMSS virulent strain and 276 in the Rahman avirulent strain; 104 of these circRNAs were shared in both strains; therefore, 329 circRNAs were virulent-specific, and 172 were avirulent-specific. A total of 20 out of 605 circRNAs originated from intergenic regions (17 were from the virulent strain, 9 were from the avirulent strain, and they shared 6 circRNAs), and 585 circRNAs were associated with a particular locus (Table 1). Notably, only 35 of the 585 circRNAs were formed by two exons.

The length distribution of *E. histolytica* circRNAs was limited, ranging from nearly 50 to 250 nucleotides (nt), peaking at 140 nt (Figure 1A), and the expression of circRNAs was practically proportional to the expression of their respective linear RNAs (Appendix A).

Despite the *E. histolytica* libraries were previously analyzed, we wanted to ensure the detection of differential expression (DE) of circRNAs between the virulent and avirulent strains. To this end, the normalized expression of circRNAs between samples was analyzed. Box plot analysis and principal components analysis (PCA) both show that the circRNAs from the three avirulent replicas have similar distribution and variation, whereas the circRNAs from one of the virulent samples are distributed slightly different and appear to have more variation than the other two samples, while separated from the avirulent circRNAs (Appendix A). The normalized expression counts were analyzed using DEseq2 library and the statistic *p* value and a fold change of 2 were determined. A volcano plot was constructed to evidence that the circRNAs were underexpressed (green dots) and overexpressed (red dots) in the *E. histolytica* virulent strain with respect to the avirulent strain (Figure 2); thus, an overexpressed circRNA in the virulent strain is underexpressed in the avirulent strain and vice-versa. Usually, circular RNA expression is lower than their linear counterparts, and for this reason, some authors prefer lower *p* values to assess DE [37]. However, in *E. histolytica*, the abundance of circRNAs expression is heterogeneous; therefore, we used two *p* values. Using *p* < 0.05, twenty DE circRNAs were identified, and twelve more were detected when a *p* < 0.1 was used. The plot shows the 32 DE circRNAs identified using both *p* values. Interestingly, three circRNAs are particularly underexpressed, all of them derived from locus EHI_169670 (a gene with three exons and two introns); one circRNA has one exon element, and the other two have two exonic elements.

Next, we used the virulent-specific and the avirulent-specific circRNAs to perform a gene ontology (GO) analysis focusing on metabolic processes. Clearly, the metabolic processes potentially regulated by the circRNAs from virulent and avirulent *E. histolytica* strains substantially differ (respectively, Figure 3A,B and Appendix A). For example, 35 GO terms were best represented in the HM1:IMSS strain, but in the Rahman strain were 47. Notably, 26 of these terms were represented in both strains, but their abundance and significance were differentially distributed. For example, taking into account the dispensability of the metabolic processes only, translational elongation, ATP metabolic processes, cellular nitrogen compound metabolic process, and gene expression appear in different X coordinates between strains. Additionally, biosynthetic processes, the generation of precursor metabolites and energy, as well as purine-containing compound metabolic processes were distributed diametrically opposed. However, as expected from metabolic processes of the same species with different virulent attributes, the term primary metabolic process showed little change between strains. This set of data suggests that the DE circRNAs might be involved in gene expression regulation in the virulent and avirulent phenotypes.

When we searched for circRNAs sponging potential, remarkably, eight of the twenty DE circRNAs were predicted to have one or more Ehi-miRNA-like target sites (Table 2), suggesting the possibility of circRNA-miRNA-mRNA regulatory mechanisms existing in *Entamoeba*. For instance, according to their prediction score, the GRIP domain RUD3 circRNAs 01_169670 and 18_169670 could potentially sponge Ehi-miR-18, Ehi-miR-136, Ehi-miR-160, Ehi-miR-50, Ehi-miR-88, and Ehi-miR-82, and Ehi-miR-17, Ehi-miR-18, Ehi-miR-160, and Ehi-miR-50, respectively.

Lastly, to verify that circRNAs are expressed in *E. histolytica,* we used RT-PCR to amplify circRNA 01_169670 (Figure 4). Sequence analysis confirmed the identity of the circRNA.

### 2.3. E. invadens circRNAs

In 20 h, encystment induced *E. invadens*, 188 circRNAs were identified, and after duplicates elimination 143 remained. All of these circRNAs were formed of single elements, where 10 of them were intergenic, 101 were exonic, and 2 were intronic (Table 1). The length distribution of *E. invadens* circRNAs was broad, ranging from nearly 130 to 710 nt, and peaked at 250 nt (Figure 1B).

Since mature cyst formation is reached after 72 h incubation in an encystment medium, the circRNAs found in trophozoites transcriptomes were also filtered, including the intergenic circRNAs whose cell-stage expression profile cannot be univocally assigned, thus 40 cyst-specific circRNAs were identified (Table 1 and Table 3). Interestingly, three circRNAs are *E. invadens*-specific (the Jacob coding EIN_023210, the cadmium metallothionein coding EIN_381500, and the high mobility protein B3 locus EIN_284560), and the rest of the circRNA-producing loci have corresponding orthologs with other amoebozoans. Three circRNAs have orthologs with the free-living *Mastigamoeba balamuthi* (EIN_061000, EIN_369710, and EIN_475930), and fifteen have more than one *E. histolytica* orthologs. Remarkably, locus EIN_461630 which codes for the splicing factor U1-70 kDa, produced three different circRNAs and has two *E. histolytica* orthologs.

Finally, we analyzed the metabolic processes GO terms of 20 h encysting *E. invadens* and their corresponding *E. histolytica* orthologs (Figure 3C). Fifty-four metabolic processes are represented in encysting amoebas indicating that distinct metabolic processes are involved during cell differentiation in *Entamoeba*. When we compared the GO terms of 20 h cysts with the virulent strain (Appendix A), again, we observed that they differed both in dispensability and distribution. For example, the metabolic process was in opposite X-axis coordinates, and the small molecule metabolic process was diametrically opposed.

## 3. Discussion

Here we bioinformatically identified circular RNA molecules from exons and intergenic regions of *E. histolytica* and *E. invadens*. Given the available suboptimal libraries used here, the number of circRNAs detected is more likely a sub-estimation. In agreement, we interpret that the lower number of circRNAs identified in *E. invadens* might not reflect the actual repertoire. For example, Mar-Aguilar and coworkers identified 199 Ehi-miRNA-like molecules [26]; however, when small RNA-specific libraries were analyzed, nearly half a million reads of 27nt small RNAs were identified in *E. histolytica* and, depending on the differentiation stage, from over 1.2 to nearly 1.9 million reads in *E. invadens* [38]. However, because of their structure, biogenesis, and half-life, we do not expect such high numbers for the circRNA repertoire.

Ninety-seven percent of the *E. histolytica* circRNAs without duplicates are exonic and constituted of a single element, and only 3% are intergenic (Table 1). We suspect that the intron mean size in the amoebic genome was the main cause for the null detection of intron-derived circRNAs. In *E. invadens*, in keeping with its gene structure, 6.99% intergenic and 1.3% intronic circRNAs were identified (Table 1). The same holds true for the distribution lengths of their respective circRNAs; *E. invadens* circRNAs double in length *E. histolytica* (Figure 1).

Mostly one circRNA per locus was identified. However, 44 loci from the virulent and 28 loci from the avirulent *E. histolytica* strains produce two or more circRNAs and up to nine circRNAs for locus EHI_169670 (Table 2 and repository). Likewise, seven loci of encystation-specific *E. invadens* also produce additional (up to five) circRNAs (Table 3). Additionally, in agreement with apparent stochastic alternative 5′ss and 3′ss selection reported in the forward-splicing of *E. histolytica* introns [35], we observed different BSJ of some circRNAs derived from a given locus. This suggests that certain circRNAs are required in larger molar quantities to achieve their roles or that they might be involved in additional metabolic tasks.

Significantly, different from the human circRNAs [39,40], the expression of *Entamoeba* circRNAs positively correlates with the expression of their linear counterparts (Appendix A). Furthermore, whereas human circRNAs originate from longer-than-average exons flanked by large introns [41,42], in both *Entamoeba* species, we observed no correlation between intron length or the number of introns per locus with the number of circRNAs produced in a given locus (Table 2 and Table 3, repository). Altogether, these data support their suggested role in gene or protein expression regulation, as has been demonstrated in the intronic flicRNAs [23,43].

As expected, we observed the DE of circRNAs between *E. histolytica* virulent and avirulent strains and detected circRNAs during *E. invadens* cell differentiation as well (Figure 3). Our data are in agreement with previous observations for amoebic small ncRNAs [38,44,45] and lncRNAs [30,46]. Interestingly, circRNA 14_116360, underexpressed in the virulent strain (i.e., overexpressed in the avirulent strain), originated from a locus that codes for a Serine-rich protein implicated in abiotic stress, whose phosphorylation might be controlled by ncRNAs [47]. Other circRNAs overexpressed in the Rahman strain correspond to loci coding for GRIP domain RUD3 proteins involved in the Golgi Apparatus structuring and trafficking (01_169670, 02_069670, 03_069670, 11_ 014170, and 18_069670) and to loci coding for the S20 and L27 ribosomal proteins (12_026410 and 16_183480, respectively). Therefore, one-quarter of the DE circRNAs (i.e., five out of seven of those overexpressed in the avirulent strain) originate from two loci coding for GRIP domain RUD3, suggesting that significant cell trafficking might be implicated in the control of core metabolic processes to maintain the avirulent phenotype, and probably in the virulent phenotype as well. This general feature reflects the differences in gene expression regulation required in distinct virulence and encystment phenotypes, encompassing all levels of control, from transcription to translation and protein modification.

Furthermore, the differences in GO terms for biological processes such as gene expression, translation elongation, ATP-metabolic process, cellular nitrogen and generation of metabolite precursors, and energy generation processes between *E. histolytica* strains and encysting *E. invadens* support this notion. The fact that some primary metabolic processes do not change between strains strengthens this view, indicating that not all biological processes are differentially regulated between the virulent and avirulent phenotypes.

Noteworthy is the encystment-specific circRNAs detected here (Table 3). As expected, and casting insights of the value of *E. invadens* as an encystment model, three E. invadens-specific circRNAs were identified. Locus EIN_023210 codes for protein Jacob, locus EIN_284560 codes for a putative high mobility group B3 protein/SWI/SNF related chromatin-binding protein, and locus EIN_381500 codes for a putative cadmium metallothionein precursor. Jacob is the most abundant glycoprotein that has chitin-binding domains and is thus an integral part of the cyst wall [48]. The fact that three loci have *Mastigamoeba balamuthi* orthologs only (Table 3) reflects the deep-branching pre-parasitic origins of encystment-related circRNAs, which may be possibly involved in encystment regulation.

When we searched for anti-Eh-miRNA target sites within the identified circRNAs, we found that eight circRNAs contain at least one and as many as six target sites for Eh-miRNAs. Probably, this number is an underestimation since we used very stringent base complementary parameters. This finding suggests a sponging function of these circRNAs, a possibility that we are currently investigating. Nevertheless, because the putative sponging circRNAs are differentially expressed in virulent amoebas (circRNAs 01_169670, 14_116360, and 18_169670 are underexpressed, and circRNAs 04_130700, 06_ DS571557, 07_036530, 19_125950, and 20_ DS571344 are overexpressed) we can envisage a circRNA-Eh-miRNA-mRNA virulence/avirulence regulatory network in which, for example, the translation of a virulence-related mRNA could be repressed by an avirulent regulatory Eh-miRNA-like molecule, and such translation repression could be relieved by the overexpression of a virulent regulatory circRNA, which will titter the avirulent regulatory Eh-miRNA (or silencing RNA molecules), and vice-versa. Recently, additional small RNA and antisense *Entamoeba* databases have been reported [38,49]. However, a statistically significant miRNA/small RNA DE comparison between strains has proved unmanageable, masking any DE relationship between circRNA and miRNA-like molecules in these strains. 

So far, regulatory networks have been predicted at the transcriptional level using the Bayesian inference [50]. Here, we propose additional regulatory networks impacting post-transcriptional expression events. Central to these regulatory networks include circRNAs that, in addition to sponging miRNA/siRNA molecules, could also interact with other ncRNAs or transcription or translation regulatory proteins and even may be translated into proteins.

In conclusion, we identified 143 and 605 reverse overlapped circRNAs from the *E. invadens* and *E. histolytica* libraries, respectively. The identified circRNAs were mostly exonic. 416 circRNAs from *E. histolytica* were virulent-specific, 267 were avirulent-specific, and 32 were shared between strains. From the latter, 32 were DE between strains, showing no correlation between intron length or the number of introns per locus with the number of circRNAs produced in a given locus. Finally, we predicted that eight of the DE circRNAs could function as sponges of the reported miRNAs in *E. histolytica*, whose functions are still unknown, although other epigenetic regulatory roles must be also explored. Our results allowed us to devise a working hypothesis to test the relationships between circRNAs and miRNA-like molecules in determining virulent/nonvirulent phenotypes and to explore additional regulatory mechanisms during amoebic encystment. Furthermore, the in silico prediction of circRNAs in this study would facilitate a deeper comprehension of regulatory epigenetic processes in such pathogenic amoebic protozoa. However, *E. histolytica* virulence is widely variable worldwide across several strains, and the relevant mechanisms controlling the differential virulent expression need to be investigated more prospectively.

## 4. Materials and Methods

### 4.1. Libraries

Total RNA *E. histolytica* RNA-Seq libraries were downloaded from the European Bioinformatics Institute, project PRJEB2766 (Available online: https://www.ebi.ac.uk/ena/browser/view/PRJEB2766?show=reads (21 September 2022)). Three libraries were from the HM1:IMSS virulent strain (113,997,619; 91,291,028; and 81,271,700 pair-end reads, respectively) and three from the Rahman avirulent strain (70,821,013; 68,278,102; and 108,168,110 pair-end reads) were used. Total RNA *E. invadens* RNA-Seq libraries were downloaded from the GenBank GEO database (Available online: htpps://www.ncbi.nlm.nih.gov/geo/query/acc.cgi?acc=GSE138449 (21 September 2022)). Four libraries from 24 h encystment induced E. invadens (34,897,385; 46,698,915; 33,531,808; and 36,965,892 pair-end reads) were used.

### 4.2. Identification of Linear and Circular RNAs

The quality control of the RNA libraries was made using the FASTQC (v0.11.8) and Trimmomatic (v0.39) software packages. To detect and quantify the linear RNAs, we improved the STAR-HTSeq pipeline. STAR software (v2.7.10a) [51] was used to make the indexed, mapping, and annotation of linear RNAs. To quantify the expression of the linear RNAs was made by the HTSeq software with the htseq-count module. circRNAs were detected using the software CIRI-full (v2.1.1) [34,52]. For the indexing and mapping of the RNA libraries, we used the program BWA (v0.7.17), as recommended by the authors of CIRI2. The quantification and annotation were made with the RO1 and RO2 modules of CIRI-full as recommended using the author’s pipeline.

### 4.3. Differential Expression of RNAs

To make the differential expression of linear RNAs and circRNAs, we used the R-base (v4.1), Rstudio IDE (v1.1.4), and the R libraries *DEseq2* and *ggplot2* to build the normalization of the expression counts, differential expression, statistical analysis, and plots.

### 4.4. Gene Ontology Analysis

GO analyses were carried out using the AmoebaDB data analysis suite. The gene ID of all detected circRNAs in both *E. histolytica* strains and *E. invadens* were uploaded and analyzed by the amoebaDB web (https://amoebadb.org) (accessed on 16 August 2022) to obtain the annotation, GO terms and GO enrichment analysis for each locus. Finally, with this data, we used the Revigo web tool (http://revigo.irb.hr) (accessed on 16 August 2022) to build the GO enrichment graph, and, with Rstudio IDE (v1.1.4), the graph was visualized.

### 4.5. Identification of circRNAs as miRNAs Sponges

To assess the prediction of circRNAs as sponges of miRNAs, we used the software intaRNA [53], which is capable of detecting RNA-RNA interactions through an alignment. The parameters used to make the predictions were a seed sequence of at least 7 nucleotides, canonical interactions between miRNA and circRNA in nucleotides 1 to 5, the possibility of gaps in the alignment, and favorable thermodynamics interaction ΔG < −7 Kcal/mol.

### 4.6. CIRI-Full Control

We designed in silico RNA decoy reads from exon end sequences of the locus EHI_110740 of *E. histolytica.* These reads were constructed by fragments of this locus to make 10 RNA library decoy reads with a length of 100 nucleotides which includes a BSJ. Additionally, we prepared the RNA libraries extracting 10% of all RNA reads, and we identified the circRNAs presented in this modified library with the described method. Next, we included the 10 artificial reads to this modified library and reidentified the circRNAs in the library.

### 4.7. RNA Isolation, Retrotranscription, and Polymerase Chain Reactions

Total RNA was isolated using the TRIzol Reagent as specified by the manufacturer (Invitrogen). For circular RNA isolation, preparations were treated with RNase R (Epicentre). Outward-facing primers targeted the second exon of *E. histolytica* locus EHI_169670 (E2as 5′CTTCTTTTTCTTTTTCTAATTCTTCACCC3´ and E2s 5′AAGAAAGTTAATGATTCTGAGAAAGAG3′) was designed as described [54] for the detection of circular RNA molecules. Retro-transcription reactions were carried out in the presence of 5 µM Actinomycin D (SigmaAldrich). The drug was added immediately after the denaturing step [55]. PCR products were amplified as described [43] and were resolved in 2.5% agarose gels.

## Figures and Tables

**Figure 1 ncrna-08-00065-f001:**
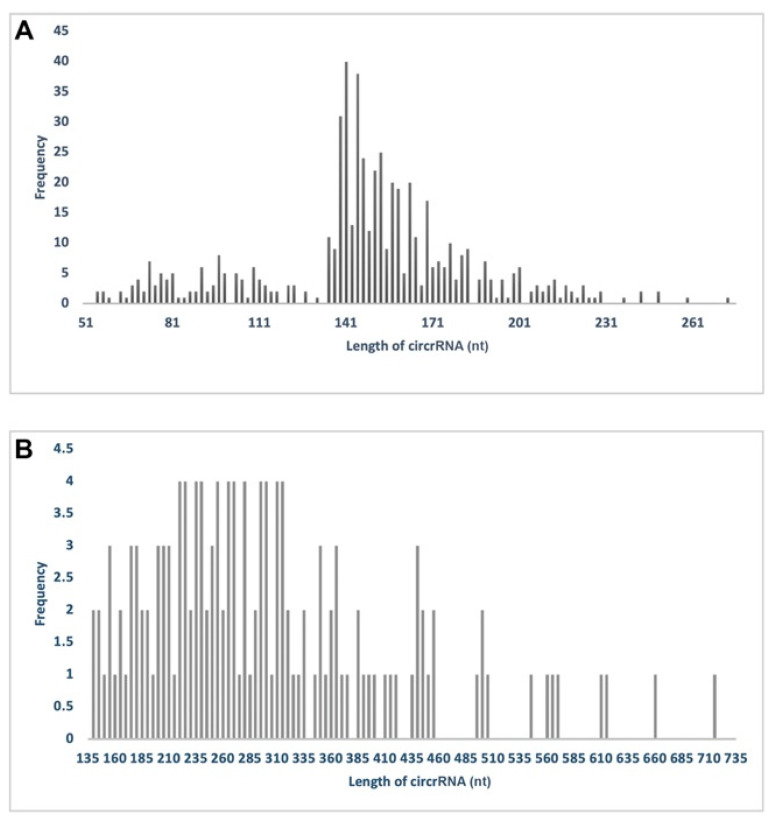
Length frequency distribution of *Entamoeba* circRNAs. The histograms show the *E. histolytica* (**A**) and *E. invadens* (**B**) circRNA length-frequency distribution.

**Figure 2 ncrna-08-00065-f002:**
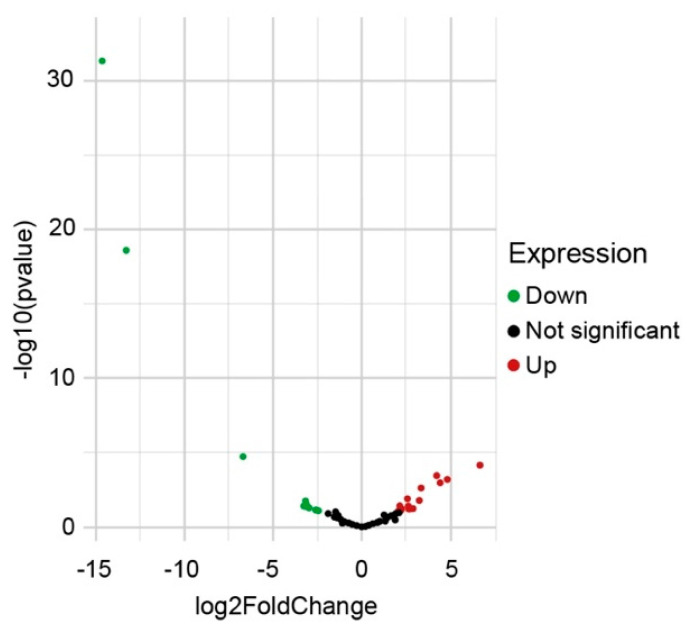
Differentially expressed *E. histolytica* circRNAs in HM1:IMSS strain with respect to Rahman strain. The volcano plot shows the DE circRNAs (*p* < 0.05 and *p* < 0.1 data are combined). 32 circRNAs were identified: 20 overexpressed (red dots) and 12 underexpressed (green dots).

**Figure 3 ncrna-08-00065-f003:**
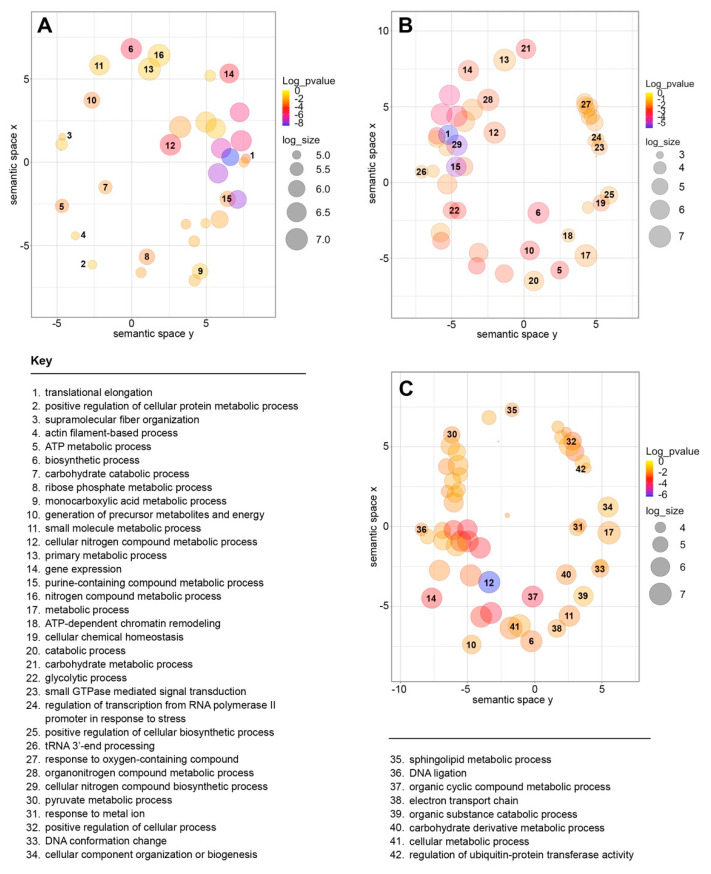
Metabolic process GO enrichment terms were identified in *E. histolytica* and *E. invadens* circRNAs. Semantic distribution for HM1:IMSS strain (**A**), Rahman strain (**B**), and 20 h encysting *E. invadens* circRNAs (**C**) are shown.

**Figure 4 ncrna-08-00065-f004:**
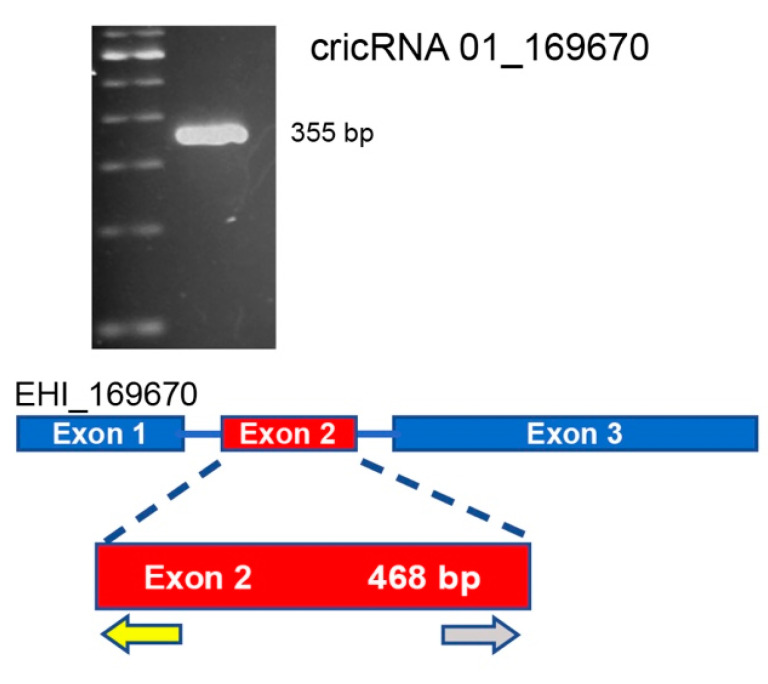
RT-PCR amplification of circRNA 01_169670. The 355 bp fragment amplified corresponds to exon 2. Arrows underneath correspond to the divergent primers used in the reaction.

**Table 1 ncrna-08-00065-t001:** Summary of Entamoeba histolytica and Entamoeba invadens circRNAs.

	*Entamoeba histolytica*	*Entamoeba invadens*
Total circRNAs	958	188
Without duplicates	605	143
mono elements	570	143
2+ elements	35	0
Exonic	585	131
Intronic	0	2
Intergenic	20	10
Virulent-specific	329	nd ^1^
Shared between virulent and avirulent	104	nd
Avirulent-specific	172	nd
Cyst-specific	nd	40

^1^ not determined.

**Table 2 ncrna-08-00065-t002:** Differentially expressed *Entamoeba histolytica* circRNAs ^1^ with predicted anti-Ehi-miRNA-like target sites.

circRNA	Element	Product	C/L ^2^	DE ^3^	Ehi-miR-like Sites In ^4^
01_169670	exon	GRIP domain RUD3	↓/↓	↓	Ehi-miR-18, Ehi-miR-136, Ehi-miR-160, Ehi-miR-50, Ehi-miR-88, Ehi-miR-82
02_169670	exon	GRIP domain RUD3	↓/↓	↓	
03_169670	exon	GRIP domain RUD3	↓/↓	↓	
04_130700	exon	Enolase	↑/⎯	↑	Ehi-miR-56
05_146110	exon	hp ^5^	↑/↑	↑	
06_DS571557	intergenic	nd ^6^	na ^6^	↑	Ehi-miR-193, Ehi-miR-39
07_036530	exon	Ribosomal S27a	↑/⎯	↑	Ehi-miR-69
08_130700	exon	Enolase	↑/⎯	↑	
09_146110	exon	hp	↑/↑	↑	
10_146370	exon	Ribosomal L27	↑/↑	↑	
11_014170	exon	GRIP domain RUD3	↓/↑	↓	
12_026410	intergenic	Ribosomal S20	↑/↓	↓	
13_086110	exon	HMG domain	↑/⎯	↑	
14_116360	exon	Serine-rich	↑/↑	↓	Ehi-miR-23, Ehi-miR-51
15_193440	exon	hp	↑/↑	↑	
16_183480	exon	Ribosomal L27	↓/↓	↓	
17_027380	exon	hp	↑/↑	↑	
18_169670	exon	GRIP domain RUD3	↓/↓	↓	Ehi-miR-17, Ehi-miR-18, Ehi-miR-160, Ehi-miR-50
19_125950	exon	ADH	↑/⎯	↑	Ehi-miR-177
20_DS571344	intergenic	nd	na	↑	Ehi-miR-150, Ehi-miR-7

^1^ The list includes those detected with *p*-value *<* 0.05 only; ^2^ C (circRNA)/L (linear RNA composed of spliced and unspliced variants) overexpressed (↑), underexpressed (↓) or unchanged (⎯); ^3^ DE, differential expression of the virulent HM1:IMSS strain with respect to the avirulent Rahman strain; ^4^ The numbers represent the Ehi-miRNA-like molecules as described by Mar-Aguilar et al. [26], possibly sponged by the corresponding circRNA, ordered according to the highest target probability; ^5^ Hypothetical protein; ^6^ Not determined/available.

**Table 3 ncrna-08-00065-t003:** *Entamoeba invadens* encystment-specific circRNAs.

20 h Cysts (#) ^1^	Gene Annotation in *E. invadens*/Gene Annotation in Orthologs ^2^	Orthologs ^2^
EIN_023210	Jacob/na ^3^	
EIN_031580	hp/hp ^4^	EHI_030890
EIN_044390	hp/hp	EHI_054800
EIN_057320	hp/C2-DCP ^5^	EHI_069320 + ^6^
EIN_061000 (3)	hp/Neuroendocrine convertase 1-like	MBAL_005770
EIN_065890	hp/hp	EHI_166920 +
EIN_065940	Sphingomyelinase C precursor/Endo- exonuclease, phosphatase-DCP	EHI_125790 +
EIN_082550 (3)	hp/Zinc finger protein	EHI_055700
EIN_092220	hp/Actin	EHI_198930
EIN_093520	ATP-dependent RNA helicase DBP6/DEAD/DEAH box helicase	EHI_052790
EIN_118080	hp/hp	EHI_016340 +
EIN_129500	60S ribosomal protein L21/60S ribosomal protein L21	EHI_069110 +
EIN_145900	DNA topoisomerase/DNA topoisomerase	EHI_120640
EIN_153430	Pyruvate phosphate dikinase/Pyruvate phosphate dikinase	EHI_009530 +
EIN_162170 (3)	hp/Deoxycytidine triphosphate deaminase	EHI_140240
EIN_176430	Vesicle-fusing ATPase/Vesicle-fusing ATPase	EHI_004640
EIN_230150 (3)	hp/WD-DCP	EHI_135040
EIN_231040	hp/hp	EHI_160980 +
EIN_274360	hp/hp	EHI_002780
EIN_281330 (3)	hp/hp	EHI_012130 +
EIN_284560	High mobility group protein B3/na	
EIN_284970	GAPDH/GAPDH	EHI_187020 +
EIN_327660	Caldesmon/Glutamic acid-rich protein precursor	EHI_182620
EIN_335410	Phospholipase D/Phospholipase D	EHI_082560 +
EIN_369450	hp/hp	EHI_152970
EIN_369520	hp/hp	EHI_091120
EIN_369710	hp/hp	MBAL_010850
EIN_377620	hp/hp	EHI_142970
EIN_381500	Cadmium metallothionein precursor/na	
EIN_398110	hp/hp	EHI_030480
EIN_403300	Myosin-2 heavy chain/Myosin heavy chain	EHI_110180
EIN_409300	60S ribosomal protein L4/60S ribosomal protein L4	EHI_000510 +
EIN_409670	hp/hp	EHI_024550
EIN_416970	Actin binding protein/Actin binding protein	EHI_094030 +
EIN_424210	dTDP-glucose 4,6-dehydratase/dTDP-glucose 4,6-dehydratase	EHI_125700 +
EIN_461630 (3)	U1-70 kDa/U1 small nuclear ribonucleoprotein subunit	EHI_153670 +
EIN_468500	Pyruvate dehydrogenase/Pyruvate:ferredoxin oxidoreductase	EHI_051060
EIN_475930	Pyrroline-5-carboxylate reductase/Pyrroline-5-carboxylate reductase	MBAL_004253
EIN_486020	hp/hp	EHI_196570
EIN_487250 (5)	Rho GTPase/Rho family GTPase	EHI_178680 +

^1^ The additional number of circRNAs in their respective loci appear between parenthesis; ^2^ As registered in the AmoebaDB for *E. invadens*/*E. histolytica* or *Mastigamoeba balamuthi*; ^3^ not available; ^4^ Hypothetical protein; ^5^ DCP (domain-containing protein); ^6^ + denotes at least one additional ortholog.

## Data Availability

The data found in this work is currently being uploaded in the AmoebaDB.

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
