# Peer review of "In Silico Identification and Characterization of circRNAs as Potential Virulence-Related miRNA/siRNA Sponges from Entamoeba histolytica and Encystment-Related circRNAs from Entamoeba invadens"

_ncrna, 2022, doi:10.3390/ncrna8050065_

Round 1

Reviewer 1 Report

To Authors,

The content of the manuscript provides advanced knowledge of non-coding RNAs in Entamoeba Biology. Due to the complicated methodology, I first suggest you generate the schematic workflow to better facilitate the understanding. After reading through it, I also would like the authors to clarify the following significant concerning issues:

1. The number of virulent-specific and avirulent-specific circRNAs in E. histolytica were 416 and 267 respectively, as informed in the text. However, in Table 1 showed 433(329) and 276(172), respectively. What is the difference between these numbers? Additionally, what do '(329)' and '(172)' represent?

2. It is also very interesting to reveal the data of virulent-specific circRNAs found in E. histolytica HM1:IMSS, especially the most expressed ones. This would contribute to the advance in the understanding of amoebic virulence. For avirulent-specific circRNAs in the Rahman strain, it would be great if discussed in comparison with HM1:IMSS. 

3. As mentioned in the column of Anti-Ehi-miR target sites, what does each number represent? Also, the authors should mention the source/reference of the reliable miR database used for mapping in this study. Just only bioinformatic evidence of miRNAs was proposed, not experimental evidence. Also, based on my experiments, it appears that miRNAs are not available in E. histolytica. It should be indeed siRNAs, not miRNAs. Please be careful of using this term because the strong evidence demonstrating the presence of miRNAs in E. histolytica is still lacking.

4. The gene annotation for each accession number should be included in Table 3 to better explain the evolution of encystment regulation among amoebic species.

5. GO enrichment analysis was less mentioned and should be more critically discussed.

6. Line 274, the term 'avirulent Eh-miRNA' is not correct because the words 'avirulent/virulent' should be referred to the pathogenicity of the strain, not for describing the biomolecules. I suggest the word 'regulatory' instead of 'avirulent'.

7. Throughout the manuscript, the scientific name, both genus, and species name, should be corrected to be italic (lines 204, 239, 242, 256, references).

8. Figure 4, please check and correct 468pb. It should be bp.

9. Some places state HM-1:IMSS, but others are HM1:IMSS without a hyphen. Please check.

In addition to the putative sponging function, what is their other possible epigenetic function? If scientifically reasonable, it could be suggested in the conclusion.

I would be more than happy to read over the revised version of this manuscript.

Cheers

Author Response

Reviewer 1 to Authors,

The content of the manuscript provides advanced knowledge of non-coding RNAs in EntamoebaBiology. Due to the complicated methodology, I first suggest you generate the schematic workflow to better facilitate the understanding.

R: As suggested, Figure S1 was added to depict the workflow and programs used.

For this reason, the previous order of the supplementary figures shifted down.

Figure S1:  Workflow for identification and partial characterization of circRNAs from Entamoeba histolytica and Entamoeba invadens, Figure S2: Design and rescue of synthetic circRNA decoy sequence from limited Entamoeba histolytica total RNA-Seq libraries, Figure S3: Near-linear correlation between circRNA and mRNA expression in Entamoeba histolytica, Figure S4: Principal Components Analysis of the six total RNA-Seq libraries used in this work,

After reading through it, I also would like the authors to clarify the following significant concerning issues:

  1. The number of virulent-specific and avirulent-specific circRNAs in  histolyticawere 416 and 267, respectively, as informed in the text. However, in Table 1 showed 433(329) and 276(172), respectively. What is the difference between these numbers? Additionally, what do '(329)' and '(172)' represent?

R: Table 1 was corrected to show virulent-specific, avirulent-specific, and shared between virulent and avirulent numbers of circRNAs. In addition, the following text was added  

Of these, 433 were found in the HM1:IMSS virulent strain and 276 in the Rahman avirulent strain; 104 of these circRNAs were shared in both strains; therefore, 329 circRNAs were virulent-specific, and 172 were avirulent-specific.

  1. It is also very interesting to reveal the data of virulent-specific circRNAs found in  histolytica HM1:IMSS, especially the most expressed ones. This would contribute to the advance in the understanding of amoebic virulence. For avirulent-specific circRNAs in the Rahman strain, it would be great if discussed in comparison with HM1:IMSS. 

R : As suggested, we discussed further the DE circRNAs

Interestingly, circRNA 14_116360, underexpressed in the virulent strain (i.e. overexpressed in the avirulent strain), originated from a locus that codes for a Serine-rich protein implicated in abiotic stress, whose phosphorylation might be controlled by ncRNAs [47]. Other circRNAs overexpressed in the Rahman strain correspond to loci coding for GRIP domain RUD3 proteins involved in the Golgi Apparatus structuring and trafficking (01_169670, 02_069670, 03_069670, 11_ 014170, and 18_069670) and to loci coding for the S20 and L27 ribosomal proteins (12_026410 and 16_183480, respectively) might be implicated in the control of core metabolic processes in this strain. This general feature reflects the differences in gene expression regulation required in distinct virulence and encystment phenotypes, encompassing all levels of control, from transcription to translation and protein modification.  

  1. As mentioned in the column of Anti-Ehi-miR target sites, what does each number represent? Also, the authors should mention the source/reference of the reliable miR database used for mapping in this study. Just only bioinformatic evidence of miRNAs was proposed, not experimental evidence. Also, based on my experiments, it appears that miRNAs are not available in  histolytica. It should be indeed siRNAs, not miRNAs. Please be careful of using this term because the strong evidence demonstrating the presence of miRNAs in E. histolyticais still lacking.

R: Table 2 was corrected to clarify the representation of the numbers appearing in the last column. Also, to attend to the reviewer’s critique and give credit to the original publication, we now use the term Ehi-miRNA-like molecules instead of Ehi-miRNA. Such terms were used throughout.

When we searched for circRNAs sponging potential, remarkably, eight of the twenty DE circRNAs were predicted to have one or more Ehi-miRNA-like target sites (Table 2),

In the results section, an explanatory text was included

For instance, according to their prediction score, the GRIP domain RUD3 circRNAs 01_169670 and 18_169670 could potentially sponge Ehi-miR-18, Ehi-miR-136, Ehi-miR-160, Ehi-miR-50, Ehi-miR-88, and Ehi-miR-82, and Ehi-miR-17, Ehi-miR-18, Ehi-miR-160, and Ehi-miR-50, respectively.

  1. The gene annotation for each accession number should be included in Table 3 to better explain the evolution of encystment regulation among amoebic species.

R: As suggested, gene annotation for each accession number was added to Table 3.

  1. GO enrichment analysis was less mentioned and should be more critically discussed.

R: as suggested, a deeper GO enrichment analysis description and discussion scription were added:

In results

For example, taking into account the dispensability of the metabolic processes only, translational elongation, ATP metabolic processes, cellular nitrogen compound metabolic process, and gene expression appear in different X coordinates between strains. Also, biosynthetic processes, the generation of precursor metabolites and energy, as well as purine-containing compound metabolic processes were distributed diametrically opposed. However, as expected from metabolic processes of the same species with different virulent attributes, the term primary metabolic process showed little change between strains. This set of data suggests that the DE circRNAs might be involved in gene expression regulation in the virulent and avirulent phenotypes.

In the discussion

Furthermore, the differences in GO terms for biological processes such as gene expression, translation elongation, ATP-metabolic process, cellular nitrogen and generation of metabolite precursors, and energy generation processes between E. histolytica strains and encysting E. invadens support this notion. The fact that some primary metabolic processes do not change between strains strengthens this view, indicating that not all biological processes are differentially regulated between the virulent and avirulent phenotypes.

  1. Line 274, the term 'avirulent Eh-miRNA' is not correct because the words 'avirulent/virulent' should be referred to the pathogenicity of the strain, not for describing the biomolecules. I suggest the word 'regulatory' instead of 'avirulent'.

R: We are grateful for the suggestion, and we incorporated such changes,

for example, the translation of a virulence-related mRNA could be repressed by an avirulent regulatory Eh-miRNA-like molecule, and such translation repression could be relieved by the overexpression of a virulent regulatory circRNA which will titter the avirulent regulatory Eh-miRNA (or silencing RNA molecules), and vice-versa.  

  1. Throughout the manuscript, the scientific name, both genus, and species name, should be corrected to be italic (lines 204, 239, 242, 256, references).

R: Scientific names were italicized throughout.

  1. Figure 4, please check and correct 468pb. It should be bp.

R: Thank you for noticing that the lower part of the figure was not translated. This issue was corrected.

  1. Some places state HM-1:IMSS, but others are HM1:IMSS without a hyphen. Please check.

R: Uniform labeling for the virulent HM1:IMSS strain was introduced.

In addition to the putative sponging function, what is their other possible epigenetic function? If scientifically reasonable, it could be suggested in the conclusion.

R: This issue was further discussed:

So far, regulatory networks have been predicted at the transcriptional level using the Bayesian inference [50]. Here we propose additional regulatory networks impacting post-transcriptional expression events. Central to these regulatory networks include circRNAs that, in addition to sponging miRNA/siRNA molecules, could also interact with other ncRNAs or transcription or translation regulatory proteins and even may be translated into proteins.

And also mentioned in the conclusion we added:

Finally, we predicted that 8 of the DE circRNAs could function as sponges of the reported miRNAs in E. histolytica, whose functions are still unknown, although other epigenetic regulatory roles must be also explored.

Reviewer 2 Report

This paper did a great job in the introduction section covering the biogenesis process of circular RNA and the current knowledge gap in the understanding of circRNA biology and function in E. histolyca and E. invaders. The authors thus mined available RNA-seq database to define circRNAs in these two species and characterized these circRNAs in regard of their distribution features and potential physiological functions. 

The manuscript will be very well-suited for Non-Coding RNA once the following comments are addressed: 

1. In Figure 2, the authors mentioned that they set two different P-value cutoffs by denoting in two different colors. It seems that the P < 0.1 blue dots are missing from this figure. 

2. In Table 2, the authors listed the predicted Ehi-miRNA targets of the given DE circRNAs. Since they proposed that there could be sponging effect between circRNA and miRNA in place, it would be interesting if the authors could go one step further and determine that whether these miRNAs are also differentially expressed in public datasets (if these datasets are available), which will further strengthen their conclusion here. 

Minor comments: 

1. Page 3, line 135, paragraph indentation 

2. Page 10, line 314, after "Rstudio IDE( v1.1.4)", there should be a "," mark. 

Author Response

The manuscript will be very well-suited for Non-Coding RNA once the following comments are addressed: 

  1. In Figure 2, the authors mentioned that they set two different P-value cutoffs by denoting in two different colors. It seems that the P < 0.1 blue dots are missing from this figure. 

R: We are sorry for the confusion. The figure now states that the plot combines both P-values and that the green dots represent underexpressed circRNAs and red dots represent overexpressed circRNAs.

  1. In Table 2, the authors listed the predicted Ehi-miRNA targets of the given DE circRNAs. Since they proposed that there could be sponging effect between circRNA and miRNA in place, it would be interesting if the authors could go one step further and determine that whether these miRNAs are also differentially expressed in public datasets (if these datasets are available), which will further strengthen their conclusion here. 

R: Unfortunately, there are no datasets for miRNAs in the different strains of Entamoeba. Some of the reported miRNAs can be inferred from the virulent strain, but the actual DE statistical analysis cannot be carried out. For this reason, we added the following paragraph to the Discussion.

R: Recently, additional small RNA and antisense Entamoeba databases have been reported [38,49]. However, a statistically significant miRNA/small RNA DE comparison between strains has proved unmanageable, masking any DE relationship between circRNA and miRNA-like molecules in these strains.  

Minor comments: 

  1. Page 3, line 135, paragraph indentation 

R: indentation was corrected, as suggested

The length distribution of E. histolytica circRNAs was limited, ranging from nearly 50 to 250 nucleotides (nt), peaking at 140 nt (Figure 1A), and the expression of circRNAs was practically proportional to the expression of their respective linear RNAs (Figure S3).

Despite the E. histolytica libraries were previously analyzed, we wanted to ensure the detection of differential expression (DE) of circRNAs between the virulent and avirulent strains. To this end, the normalized expression of circRNAs between samples was analyzed. Box plot analysis and principal components analysis (PCA) both show that the

  1. Page 10, line 314, after "Rstudio IDE( v1.1.4)", there should be a "," mark. 

R: punctuation was corrected

Rstudio IDE (v1.1.4), and the R libraries DEseq2 and ggplot2 to build the normalization of the expression counts

Round 2

Reviewer 1 Report

To authors,

The revised manuscript has been much improved and more comprehensive. However, as mentioned regarding the evidence of miRNA availability in E. histolytica, the research title should be not specific only to miRNA, preferably to substitute with 'the potential virulence-related miRNA/siRNA sponges'.

Also, in the abstract, it is better to state specifically in Line 34 and Line 36 with 'bioinformatically reported miRNAs', and 'to test the possible circRNAs/miRNAs/siRNAs interactions', respectively.

Line 90: the word 'predicted' is repetitious, so delete the other one,

Line 120: omit the outer bracket ERR058010)

Table 2: the annotation 'Anti-Ehi-miRNA-like sites in' written in the last column is quite confusing. So, this column description should be replaced with 'Ehi-miRNA-like target sites in', easier to understand than the used term.

Table 3: To be more informative, the annotation of orthologs in E. histolytica should be described also.

To raise the impact of this paper, I would suggest the authors more to emphasize that 'The silico prediction of circRNAs in this study would facilitate the deeper comprehension of regulatory epigenetic processes in such pathogenic amoebic protozoa. However, E. histolytica virulence is widely variable worldwide across several strains and the relevant mechanisms controlling the differential virulent expression need to be more investigated prospectively.'

Congratulations in advance!

Author Response

Art1 LLMA Comments and Suggestions for Authors

Round 2, reviewer 1

The revised manuscript has been much improved and more comprehensive. However, as mentioned regarding the evidence of miRNA availability in E. histolytica, the research title should be not specific only to miRNA, preferably to substitute with 'the potential virulence-related miRNA/siRNA sponges'.

R: as suggested the title was changed to

In silico identification and characterization of circRNAs as potential virulence-related miRNA/siRNA sponges from Entamoeba histolytica andencystment-related circRNAs from Entamoeba invadens.

Also, in the abstract, it is better to state specifically in Line 34 and Line 36 with 'bioinformatically reported miRNAs', and 'to test the possible circRNAs/miRNAs/siRNAs interactions', respectively.

R: both lines were corrected as suggested.

Out of the common circRNAs, 32 were DE between strains. Finally, we predicted that 8 of the DE circRNAs could function as sponges of the bioinformatically reported miRNAs in E. histolytica, whose functions are still unknown. Our results extend the E. histolytica RNAome and allow us to devise a hypothesis to test circRNAs/miRNAs/siRNAs interactions in determining the virulent/nonvirulent phenotypes and to explore other regulatory mechanisms during amoebic encystment.

Line 90: the word 'predicted' is repetitious, so delete the other one,

R: the second ‘predicted’ word was deleted.

confirmed the presence of microRNA-like molecules that had been predicted bioinformatically in E. histolytica [27].

Line 120: omit the outer bracket ERR058010)

R: The outer bracket was deleted out; thank you for noticing the typo.

six E. histolytica (run access codes ERR058005 – ERR058007 for the virulent HM1:IMSS strain and ERR058008 – ERR058010 for the avirulent Rahman strain)

Table 2: the annotation 'Anti-Ehi-miRNA-like sites in' written in the last column is quite confusing. So, this column description should be replaced with 'Ehi-miRNA-like target sites in', easier to understand than the used term.

R: the column heading was edited as suggested.

circRNA

Element

Product

 C / L 2

 DE 3

Ehi-miR-like sites in 4

01_169670

exon

GRIP domain RUD3

↓ / ↓

Ehi-miR-18, Ehi-miR-136, Ehi-miR-160, Ehi-miR-50, Ehi-miR-88, Ehi-miR-82

Table 3: To be more informative, the annotation of orthologs in E. histolytica should be described also.

R: The annotation of E. invadens, E. histolytica, and M. balamuthi were included in Table 3.

20 h cysts (#) 1

Gene annotation in E. invadens / Gene annotation in orthologs 2

Orthologs 2

EIN_023210

Jacob / na 3

EIN_031580

 hp / hp 4

EHI_030890

EIN_044390

hp / hp

EHI_054800

EIN_057320

hp / C2-DCP 5

EHI_069320 + 6

EIN_061000 (3)

hp / Neuroendocrine convertase 1-like

MBAL_005770

EIN_065890

hp / hp

EHI_166920 +

EIN_065940

Sphingomyelinase C precursor / Endo- exonuclease, phosphatase-DCP

EHI_125790 +

EIN_082550 (3)

hp / Zinc finger protein

EHI_055700

EIN_092220

hp / Actin

EHI_198930

EIN_093520

ATP-dependent RNA helicase DBP6 / DEAD/DEAH box helicase

EHI_052790

EIN_118080

hp / hp

EHI_016340 +

EIN_129500

60S ribosomal protein L21 / 60S ribosomal protein L21

EHI_069110 +

EIN_145900

DNA topoisomerase / DNA topoisomerase

EHI_120640

EIN_153430

Pyruvate phosphate dikinase / Pyruvate phosphate dikinase

EHI_009530 +

EIN_162170 (3)

hp / Deoxycytidine triphosphate deaminase

EHI_140240

EIN_176430

Vesicle-fusing ATPase / Vesicle-fusing ATPase

EHI_004640

EIN_230150 (3)

hp / WD-DCP

EHI_135040

EIN_231040

hp / hp

EHI_160980 +

EIN_274360

hp / hp

EHI_002780

EIN_281330 (3)

hp / hp

EHI_012130 +

EIN_284560

High mobility group protein B3 / na

EIN_284970

GAPDH / GAPDH

EHI_187020 +

EIN_327660

Caldesmon / Glutamic acid-rich protein precursor

EHI_182620

EIN_335410

Phospholipase D / Phospholipase D

EHI_082560 +

EIN_369450

hp / hp

EHI_152970

EIN_369520

hp / hp

EHI_091120

EIN_369710

hp / hp

MBAL_010850

EIN_377620

hp / hp

EHI_142970

EIN_381500

Cadmium metallothionein precursor / na

EIN_398110

hp / hp

EHI_030480

EIN_403300

                     Myosin-2 heavy chain / Myosin heavy chain

EHI_110180

EIN_409300

60S ribosomal protein L4 / 60S ribosomal protein L4

EHI_000510 +

EIN_409670

hp / hp

EHI_024550

EIN_416970

Actin binding protein / Actin binding protein

EHI_094030 +

EIN_424210

dTDP-glucose 4,6-dehydratase / dTDP-glucose 4,6-dehydratase

EHI_125700 +

EIN_461630 (3)

U1-70 kDa / U1 small nuclear ribonucleoprotein subunit

EHI_153670 +

EIN_468500

Pyruvate dehydrogenase / Pyruvate:ferredoxin oxidoreductase

EHI_051060

EIN_475930

Pyrroline-5-carboxylate reductase / Pyrroline-5-carboxylate reductase

MBAL_004253

EIN_486020

hp / hp

EHI_196570

EIN_487250 (5)

Rho GTPase / Rho family GTPase

EHI_178680 +

1 The additional number of circRNAs in their respective loci appear between parenthesis; 2 As registered in the AmoebaDB for E. invadens/E. histolytica or Mastigamoeba balamuthi ; 3 not available; 4 Hypothetical protein; 5 DCP (domain-containing protein); 6 + denotes at least one additional ortholog.

To raise the impact of this paper, I would suggest the authors more to emphasize that 'The in silico prediction of circRNAs in this study would facilitate the deeper comprehension of regulatory epigenetic processes in such pathogenic amoebic protozoa. However, E. histolytica virulence is widely variable worldwide across several strains, and the relevant mechanisms controlling the differential virulent expression need to be more investigated prospectively.'

R: the text was added to the conclusion as suggested.

Our results allowed us to devise a working hypothesis to test the relationships between circRNAs and miRNA-like molecules in determining virulent/nonvirulent phenotypes and to explore additional regulatory mechanisms during amoebic encystment. Furthermore, the in silico prediction of circRNAs in this study would facilitate a deeper comprehension of regulatory epigenetic processes in such pathogenic amoebic protozoa. However, E. histolytica virulence is widely variable worldwide across several strains, and the relevant mechanisms controlling the differential virulent expression need to be investigated more prospectively.

Round 3

Reviewer 1 Report

Congratulations on your job well done!